# Carbon limitation in hypereutrophic, periphytic algal wastewater treatment systems

**Brandon J. Furnish[¤], Troy A. Keller** [ID]*

Department of Earth and Space Sciences, Columbus State University, Columbus, Georgia, United States of America

¤ Current address: Department of Engineering, Liberty Utilities, Columbus, Georgia, United States of America
* keller_troy@columbusstate.edu

**Data Availability Statement:** All data files are available from the CSU Press Database accessible at (https://csuepress.columbusstate.edu/datasets/2/).

## Abstract

Global eutrophication degrades water quality in freshwater ecosystems and limits the availability of freshwater for human consumption. While current wastewater treatment facilities (WWTF) remove pathogens and pollutants, many US WWTF continue to discharge nutrients that contribute to eutrophication. Traditional nutrient removal technologies can effectively reduce eutrophication risk, but can have unintended negative consequences on human and environmental health. Alternatives, such as algae-based treatment systems, improve the sustainability of the nutrient recovery process by producing biomass that can be converted to biofuel. However, research is needed to increase the productivity of algal treatment systems to improve their economic viability. Because algae in wastewater treatment systems are grown in wastewater rich in nutrients, the algae could become limited by dissolved inorganic carbon. This hypothesis was tested in 1.2 m long recirculating floways (n = 8 for each treatment/control) by quantifying algal dry mass and wastewater nutrient concentrations in 3 independent experiments: (1) carbon dioxide gas infused vs air infused control; (2) hydrochloric acid acidified vs neutralized solution control vs no chemical addition control; and (3) sodium bicarbonate addition vs no chemical control. Results showed increases in algal biomass after 18 days in wastewater augmented with dissolved inorganic carbon (carbon dioxide or sodium bicarbonate). In contrast, maintaining wastewater at near neutral pH with hydrochloric acid reduced algal productivity relative to controls. Nutrient reductions generally paralleled algal biomass increases except in the bicarbonate addition experiment. These findings provide evidence for the importance of carbon limitation in algal wastewater treatment floways. These results could help explain why carbon dioxide infusions stimulate algae in treatment systems. Furthermore, these results suggest that algae in nutrient enriched, sun-exposed streams (e.g., agricultural ditches or urbanized streams) may become carbon limited during peak periods of productivity. These findings could have important implications for ecosystems undergoing eutrophication as atmospheric carbon dioxide concentrations continue to rise.

**Funding:** This research was partially funded by Columbus State University's SRACE grant [BJF], Liberty Utilities (BJF), and the Department of Earth and Space Sciences [TAK]. The funding groups did not influence the research conclusions or its publication. Liberty Utilities provided support in the form of a salary for one author [BJF], but this entity did not have any additional role in the study design, data collection and analysis, decision to publish, or preparation of the manuscript. The specific roles of the authors are articulated in the 'author contributions' section.

**Competing interests:** Liberty Utilities is a commercial entity that employed and paid the salary of one author [BJF] during the final preparation of this manuscript. This does not alter our adherence to PLOS ONE policies on sharing data and materials. The authors declare that no competing interests exist.

## Introduction

Cultural eutrophication, anthropogenic nutrient over-enrichment of natural waters, has degraded aquatic ecosystems globally [1]. Phosphorus, a limiting nutrient often involved in eutrophication, has increased by 75% globally compared to preindustrial levels in terrestrial and freshwater ecosystems [2]. This phosphorus loading translates into ~13 Tg/yr of phosphorus accumulating in freshwaters and surface soils, ultimately effecting marine environments [2]. Nutrient over-enrichment can cause algal blooms, reduce water quality, and decrease aquatic biodiversity [3]. While evidence suggests that progress toward reducing nutrients has been made in some areas, additional actions are needed to lower nutrients in US rivers more generally [4].

Surface waters can become enriched in nutrients when they receive discharges of domestic sewage, effluent from industrial production, or runoff from agricultural fields and impervious surfaces [5]. To protect the quality of surface waters, EPA mandated that total maximum daily loads be set to limit nutrients, sediments, and effluent temperatures, for all impaired state waters [6]. Additionally, the Clean Water Act mandates that wastewater treatment facilities (i.e., point sources) have a permit to discharge nutrients, chemicals, and biological material from their effluent [7].

Municipal wastewater treatment facilities were designed to treat water from sewer lines connected to households, businesses, and factories. Most wastewater treatment facilities operate a two-stage process referred to as primary and secondary treatment [8]. Primary and secondary treatment processes typically ensure nutrient levels in effluent meet federal and state regulations, but these technologies show only 78% phosphorus removal efficiency for a key cause of cultural eutrophication, total phosphorus [9]. To further reduce nutrient concentrations in discharges, some water treatment facilities have expanded their treatment process to include a tertiary stage such as biological nitrogen removal [10]. The adoption of advanced nutrient removal technologies can reduce the risk of eutrophication, but a lifecycle analysis revealed that these technologies can have unintended negative consequences for human health and environmental integrity [11].

Due to their affinity for nutrients, algae can be incorporated into wastewater treatment systems to serve as a sustainable alternative to typical tertiary nutrient removal technologies [12, 13]. One key advantage of algae is that they can lower concentrations nitrogen and phosphorus down to in wastewater than chemical treatments [14]. For instance, when algae were used to treat wastewater the removal efficiency of phosphorus was as high as 97% [15]. Algal treatment systems also sequester atmospheric-derived carbon dioxide [16]. Furthermore, the sustainability of algal treatment systems is improved when waste algal biomass is converted to renewable biofuels. Creating biofuels from waste algae can reduce carbon emissions [16] and enhance economic feasibility [17]. While there are several different algal treatment system configurations, algal turf scrubbers™, the focus of this research, remove nutrients using attached filamentous algae. These algal treatment systems have high productivity, show effective nutrient removal, are easily harvested and dewatered, and produce copious biomass for biofuels [18].

Atmospheric carbon dioxide diffuses into water and serves as a source of dissolved inorganic carbon required for algal photosynthesis [19]. As algae remove dissolved inorganic carbon, more carbon dioxide from the atmosphere diffuses into the water to re-establish the air-water equilibrium. Although atmospheric carbon dioxide readily diffuses into water [20], algae may become carbon limited in highly productive algal treatment systems [21, 22] when algae require more carbon than diffusion can supply. Carbon limitation could explain why dairy wastewater treated using algal turf scrubbers™ augmented with carbon dioxide effluent showed

enhanced algal biomass [23]. It may also explain why carbon dioxide stimulated algal productivity in wastewater high rate algal ponds [24, 25] and lab trials with synthetic wastewater [26].

During periods of rapid photosynthesis, such as those found on algal turf scrubbers, algae can raise pH [27]. High pH values can negatively affect algal growth and productivity [28, 29]. Azov [30] regulated the pH of wastewater using carbon dioxide and found that the planktonic alga *Scenedesmus obliquus* had its highest productivity when carbon dioxide was used to maintain a near neutral pH of 7.5.

Because carbon dioxide infusions simultaneously lower pH and raise DIC, the mechanism resulting in increased algal productivity has not often been assessed. Cole et al. [31] augmented DIC in high rate algal ponds. The study's results were consistent with the carbon limitation hypothesis because experiments showed an 87% and 89% increase in *Oedogonium intermedium* biomass when molasses and carbon dioxide (respectively) were used as carbon sources.

While the benefits of adding carbon dioxide to algal wastewater treatment systems have been established [29, 32–34], questions remain about the mechanism by which carbon dioxide improves productivity (i.e., reducing carbon limitation or suppressing elevated pH). The goal of this study was to determine if carbon dioxide stimulated algal production in algal treatment systems and if so, by what mechanism (i.e., pH regulation versus DIC augmentation). The study implemented a strong inference-type hypothesis framework [35] using a series of three independent experiments. The first experiment infused carbon dioxide into wastewater to assess the importance of carbon limitation/pH regulation in filamentous algal wastewater treatment systems. In a second experiment pH was maintained at near neutral levels using carbon-free acid to evaluate how pH effects system productivity and treatment efficiency. A final experiment evaluated algal growth and nutrient removal capacity after the addition of a different DIC source, sodium bicarbonate. We hypothesized that carbon limitation is the primary mechanism controlling algal productivity in these highly productive filamentous algal wastewater treatment systems.

## Materials and methods

### General experimental conditions

To conduct the carbon infusion experiments, replicated experimental wastewater treatment floways were constructed of cylindrical polyvinyl chloride pipes (PVC: 61 cm x 121 cm x 5.1 cm, height x length x inside pipe diameter, Fig 1). A 7.62 cm wide opening was cut into the top of each floway so that the wastewater received light from grow lights placed tangential to the direction of flow. Each flume was filled with approximately 7 L of wastewater collected from a clarifier at Columbus Water Works Wastewater Treatment Facility in Columbus, Georgia (USA). The wastewater was collected on the start date of each experiment. Thirty-two (2.5 cm x 2.5 cm) unglazed ceramic tiles were placed inside the floways as substrates for periphytic algae. Each flume was fashioned with a flow-controlled air-line and aeration stone to circulate the wastewater (Fig 1). To establish uniform flow conditions, air injection rates were adjusted until a 3 cm x 0.5 cm x 0.01 cm (height x length x thickness) piece of polyethylene sheeting submerged 2 cm in the circulating wastewater was deflected approximately 45 degrees. Flume water levels were monitored daily, and Milli-Q® Ultrapure water was added to maintain a constant volume of water without introducing additional nutrient or ions. Light for photosynthesis was provided using 6 1.2 m x 38.1 mm (length x diameter) GE™ F40 T12 grow lights which produced a color appearance of 3100K at 1900 Lumens. The lights were set on a 12-hour time interval to simulate night and day and were placed on the floways tangential to the direction of flow. Instead of using algal seed stock, algae in the wastewater was allowed to colonize and grow naturally.

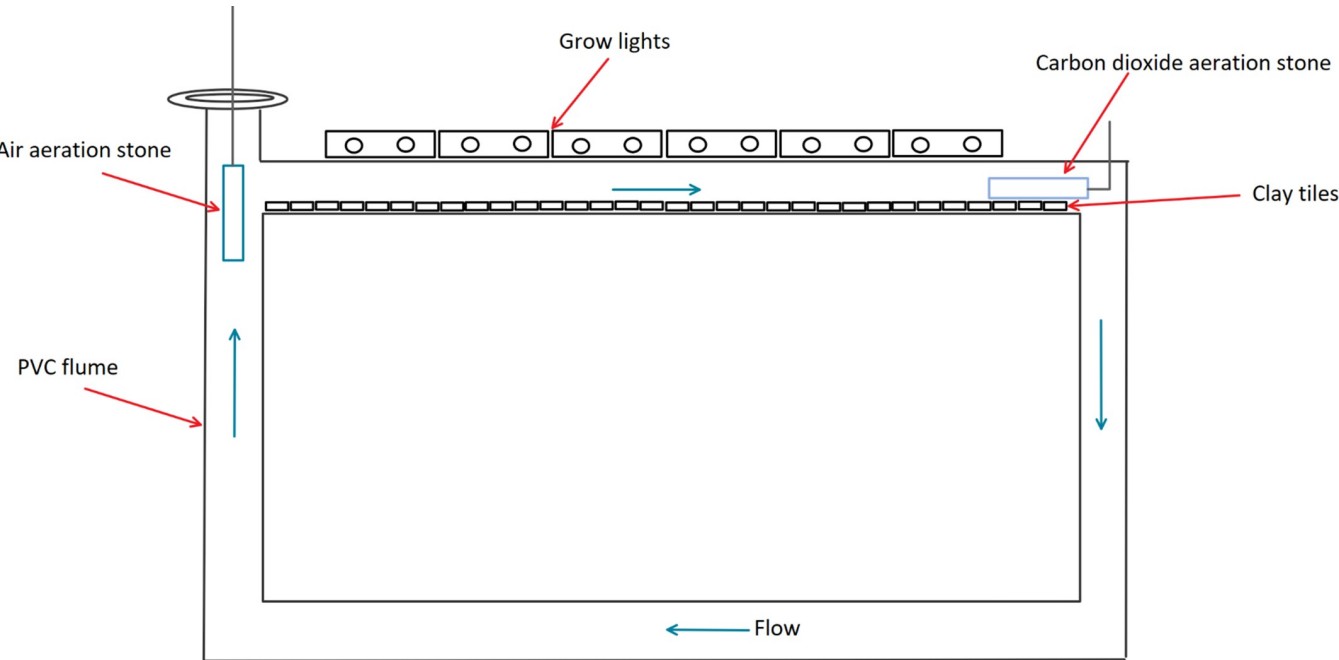

**Fig 1. Diagram of a recirculating floway used in the carbon dioxide experiment.** Polyvinyl chloride pipe (PVC) floways were filled with secondarily treated wastewater. Unglazed clay were placed in the upper part of the floways for algal substrate and illuminated with fluorescent grow lights. Compressed air was diffused into the floways to induce circulation. Arrows indicate the direction of flow.

## Physicochemical measurements

To characterize nutrient conditions, 80 mL of water were collected from each floway, placed in Whirl-Paks™, and preserved by adding one drop of concentrated sulfuric acid (36N). Milli-Q® water and Hach® effluent wastewater standards were preserved in the same manner and used as method blanks and standard controls. Samples were stored at 4˚C until analysis, which was no longer than 30 days after collection.

Directly before analysis, preserved wastewater samples were raised to 25˚C and to pH 7 using 0.1 N sodium hydroxide. For nitrate analysis, samples were analyzed using a cadmium reduction method [Method 8039, 36]. Orthophosphate samples were analyzed using a molybdate ascorbic acid colorimetric reaction [Method 8048, 37]. Total phosphorus was analyzed after acid persulfate digestion using a molybdate ascorbic acid colorimetric analysis of resulting orthophosphate [Method 8190, 38]. Total nitrogen was analyzed using an alkaline persulfate digestion followed by nitrate analysis using chromotropic acid reaction [Method 10071, 39]. A Hach® DRB200 block was used to heat test tubes for the total phosphorus (150˚C, 30min) and total nitrogen digestion period (105˚C, 30min). The concentration of all nutrient species was quantified using a Hach® DR 2700 spectrophotometer. For quality assurance and control, a Hach® wastewater standard was used to determine the accuracy of the procedures.

The temperature and pH of each flume's wastewater was recorded daily using a Hach® EC10 which was recalibrated when readings differed from pH 7 standard by more than 0.02. The probe was calibrated using Hach® pH standard solutions 4.01, 7.00, and 10.01. Light intensity was measured at the beginning, middle, and end of each floway's opening using a Li-Cor® Quantum LI-190R sensor and LI-1400 Data Logger.

On the last day of each experiment, three random rows of tiles were selected from each floway, one was analyzed for dry mass, volatile solids, and chlorophyll. Samples were stored frozen (-4˚C) in Whirl-Paks™ prior to analysis, which was no longer than 30 days after collection.

## Quantifying algal biomass and species composition

To prepare filters for dry mass and volatile solids measurements, pre-rinsed glass fiber filters (GF/F, 0.7 μm, Wyvern Scientific) were placed in numbered, pre-weighed aluminum weigh pans, ignited at 550°C for 15 minutes in a muffle furnace, and cooled in Drierite™ filled desiccators before being weighed to the nearest 0.1mg. Drierite™ was placed inside the scale chamber to minimize moisture wicking during weighing. Algae were rinsed and scraped off tiles onto filters using Milli-Q® water and a Kartell™ spatula before being dried in an oven for 24h at 105°C. Dry mass was determined after filters were cooled in Drierite™ filled desiccators and weighed (± 0.1 mg). To estimate ash-mass, filters with dry algae were then ignited at 550°C for 30 min. in a muffle furnace, cooled in desiccators, and weighed to the nearest 0. 1 mg using a Mettler Toledo AT400 Precision Digital Balance. To calculate the volatile solids, the weight of the ashed algae was subtracted from the dry weight of the algae [Method 150.1, 40].

To examine species composition, algae samples were collected from random tile locations on days 9 and 18 from each flume and observed using a Leica DM500 microscope at 400x magnification.

**Experiment 1: Carbon dioxide infusion.**   To characterize the effects of carbon dioxide infusion on nutrient removal and algal growth, wastewater in half of the floways were infused with 100% food grade gaseous carbon dioxide. Eight floways, selected using a random number generator, were administered carbon dioxide using an aeration stone submerged downstream of the tiles. Eight other floways were left untreated as controls. The experiment was conducted indoors for 18 days from Sept. 10, 2016 through Sept. 28, 2016.

**Experiment 2: pH manipulation.**   Because carbon dioxide infusions reduce pH as well as increase dissolved inorganic carbon, a second experiment was designed to determine if pH influences nutrient uptake and algal biomass production. Eight floways, selected at random, were administered 0.5 N HCl daily to maintain an average pH of 6.4 (n = 8). This pH was calculated to be the average pH of the eight floways infused with carbon dioxide in experiment 1. To control for the addition of chloride when adding HCl to the pH treatment floways, another eight floways, selected at random, were administered a neutralized solution (pH = 7) of sodium hydroxide (NaOH) and HCl daily (n = 8). The volume of the pH neutral solution added daily was calculated as the average volume of HCl added to the eight floways in order to maintain a pH of 6.4. An additional eight floways were left untreated as experimental controls (n = 8). The experiment was conducted from March 12, 2017 to March 29, 2017. Volatile solids were excluded from this experiment's analyses because results indicated problems with moisture wicking that made the measurements unreliable.

**Experiment 3: Sodium bicarbonate addition.**   The sodium bicarbonate addition experiment was designed to investigate the efficacy of adding an alternative carbon source to wastewater other than gaseous carbon dioxide. Eight floways, selected at random, were administered sodium bicarbonate to create a starting concentration of 1.14 g/L (0.014 M). Eight floways were used as unmanipulated controls. The experiment was conducted from July 22, 2017 to August 8, 2017.

Alkalinity, a measure of water's ability to resist changes in pH, is often controlled by the bicarbonate buffer system. Since most of the dissolved inorganic carbon in water is part of the buffer system, alkalinity can provide an indicator of carbon content, particularly when strong bases are absent [41]. This measurement was used to determine how much, if any, sodium bicarbonate needed to be added to treated floways to maintain the target dissolved inorganic carbon concentrations. Alkalinity of all floways was measured on days 1, 9, and 18 by titrating 1.6N sulfuric acid into the sample and measuring the volume required to achieve a pH = 4.5. Based on White et al. 2013 [42], the alkalinity of sodium bicarbonate treated flumes was also

measured on day 2 to check that the target alkalinity (800 mg/L as calcium carbonate) was achieved.

## Statistical analysis

In several instances, assessing the performance of experimental treatments required that repeated measurements be taken from each replicate throughout each experiment. Multiple measures of the same floway violate the assumption of independence required for the use of standard analysis of variance models. Repeated measures analysis of variance is a statistical model that accounts for this lack of independence. Thus, repeated measures analysis of variance (RM) served as an appropriate model to assess treatment effects when one dependent variable was measured on two or more occasions (e.g., nitrate, orthophosphate, total nitrogen, and total phosphorus). The repeated measures analysis of variance model assumes that the data show equal variances of the differences among all pairs of treatments (i.e., sphericity). Deviations from sphericity can inflate Type 1 error and reduce power [43]. Mauchly's test of sphericity was used to assess sphericity; Greenhouse-Geisser corrected significance values were reported for all within-subjects RM effects (i.e., time and time*treatment) because some comparisons failed to meet the assumption of sphericity [44]. Nitrate and orthophosphate concentrations were analyzed for days 1, 9, and 18. Total nitrogen and total phosphorus concentrations were analyzed for days 1 and 18 for the carbon dioxide infusion experiment and days 1, 9, and 18 for the pH and sodium bicarbonate experiments. Because dry mass and volatile solids were measured only on the last day of the experiments, a one-way analysis of variance (AOV) was used to assess treatment effects. Similarly, initial pH and temperature conditions were compared among the three experiments using one-way AOV. Initial nutrient concentrations were analyzed using multiple analysis of variance (MAOV) followed by univariate AOVs. To determine where differences occurred between groups, pairwise post hoc comparisons were made using the Tukey HSD (Tukey) or Bonferroni (Bonf) since both corrected for the number of pairwise comparisons [45]. Homogeneity of variances were evaluated using Levene's test of equality of error variances. Alpha was set to 0.05. All statistical analyses were performed using IBM SPSS 26.0 [46].

## Results

### General experimental conditions

To compare baseline testing conditions across experiments, initial samples were collected and analyzed after wastewater was distributed to the floways. Initial temperatures of the wastewater differed among experiments (Fig 2, AOV, p < 0.001). The carbon dioxide infusion and sodium bicarbonate addition experiments had similar average temperatures 25.1 ± 0.1˚C (mean ± 1SD) and 25.2 ± 0.2˚C respectively (Tukey, p = 0.805). However, the temperature of water used in the pH variable experiment averaged 20.1 ± 0.5˚C, significantly lower compared to the carbon dioxide or sodium bicarbonate experiments (Fig 2, Tukey, p < 0.001 for all).

Initial wastewater pH was circumneutral however detectable differences existed among experiments (AOV p < 0.001). The pH experiment's wastewater had a higher initial average pH 7.6 ± 0.20 than wastewater from the carbon dioxide infusion experiment and sodium bicarbonate addition 6.7 ± 0.82 and 6.7 ± 0.43 respectively (Fig 2B, Tukey, p < 0.001 for all).

Likewise, nutrient concentrations varied among experiments (MAOV p < 0.001). Initial nitrate, orthophosphate, total phosphorus and total nitrogen concentrations differed across all experiments (Fig 3, AOV, p < 0.001 for all). Initial nitrate concentrations were lower in the bicarbonate (55% lower) and carbon dioxide experiment (26% lower, Tukey, p < 0.001 for all) compared to the pH experiment (4.4 ± 0.41 N mg/L). On average, nitrate composed 42% of the total nitrogen in the wastewater among all experiments (Fig 3). The sodium bicarbonate

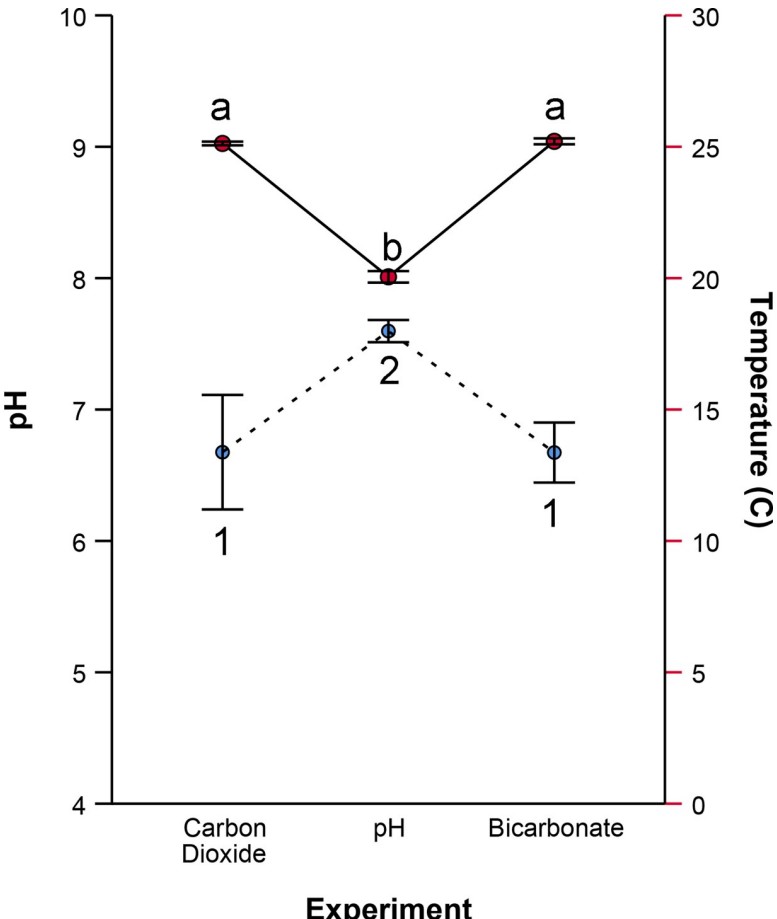

**Fig 2. Initial environmental conditions among experiments.** Points represent average initial temperature (solid line) and pH (dashed line) for the carbon dioxide (n = 16), pH (n = 24), and bicarbonate experiments (n = 16). Error bars indicate 95% confidence intervals. Points labeled with similar letters were compared statistically and those bearing unique numbers differed statistically in pairwise, post hoc comparisons (Tukey p < 0.05 for all significant comparisons).

experiment had significantly greater initial orthophosphate concentrations (1.4 ± 0.12 P mg/L) than either the carbon dioxide (23% lower) or pH experiments (42% lower, Tukey, p < 0.005 for both). The pH experiment had 25% lower initial orthophosphate concentrations compared to the carbonate experiment (Fig 3, Tukey, p = 0.012). Overall average orthophosphate composed only 18% of total phosphorus among all experiments.

The experimental floways were dominated by filamentous green algae in each experiment. The two most abundant taxa were always *Ulothrix zonata*, and *Oedogonium* sp. Diatoms were also present but were far less abundant. These experiments were conducted indoors to minimize the colonization by adult Chironomids. While a few larval midges were observed, they were not abundant. Thus, our observations indicated that the experimental floways were composed predominately of primary producers (mostly green algae) without grazers or predators.

## Experiment 1: Carbon dioxide infusion

The effectiveness of infusing wastewater with gaseous carbon dioxide was determined by comparing algal biomass between treatments (with carbon dioxide infusions) and controls (no

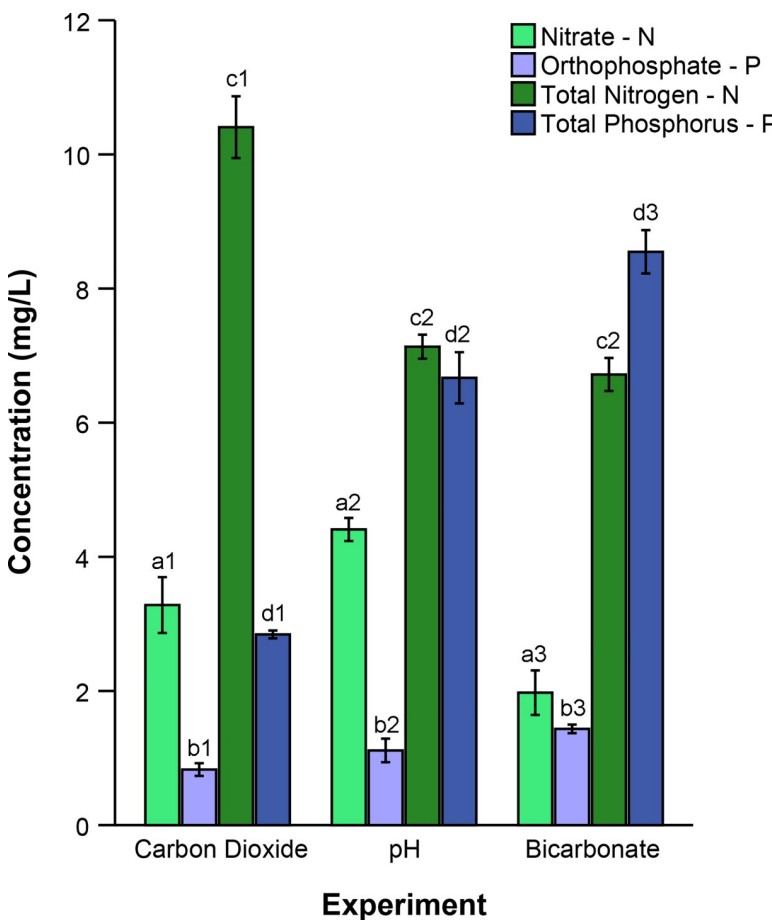

**Fig 3. Initial nutrient concentrations in wastewater used in the experiments.** Average nitrate (light green), orthophosphate (light blue), total nitrogen (dark green) and total phosphorus (dark blue) concentrations (mg/L) were calculated for the carbon dioxide (n = 16), pH (n = 24), and bicarbonate experiments (n = 16). Error bars indicate 95% confidence intervals. Bars labeled with similar letters were compared statistically and those bearing unique numbers differed statistically in pairwise, post hoc comparisons (Tukey p < 0.05 for all significant comparisons).

infusion). Floways infused with carbon dioxide averaged 66% greater algal dry mass compared to controls (Fig 4, AOV, p = 0.002). On average, the algae showed volatile solids averaging 87% of total mass (Fig 4).

To further evaluate the effects of carbon limitation in algal floways, nutrient concentrations were compared across treatments. Carbon dioxide treated floways had a significantly lower nitrate concentration averaging 24% less than controls (Fig 5A, RM, p = 0.011). Treated floways averaged 22% lower orthophosphate concentrations versus controls (Fig 5B, RM, p = 0.005). There existed no statistically significant difference between treated and control floways for total nitrogen (Fig 5C, RM, p = 0.818). Total nitrogen concentrations were only 1% lower in treated floways versus controls. In contrast, total phosphorus concentrations were 11% lower in treated floways relative to controls (Fig 5D, RM, p = 0.001).

Nitrate, orthophosphate, total nitrogen, and total phosphorus declined significantly over the course of the experiment (Fig 5, RM, p < 0.001 for all). Nitrate was reduced by 96% over the course of the experiment. Most of that decline in nitrate (58%) occurred from day 1 to 9 with the remainder occurring from day 9 to 18 (Fig 5A, Bonf, p < 0.001 for both). Over the course of the experiment, orthophosphate concentrations declined an average of 77%. Similar

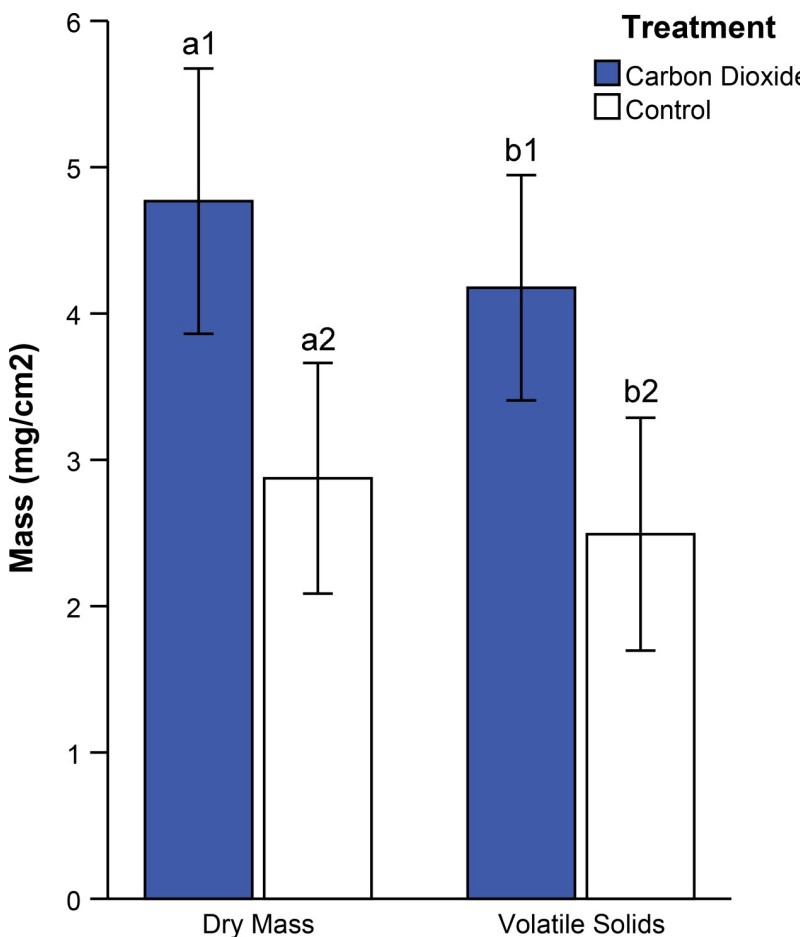

**Fig 4. Effects of carbon dioxide on algal mass.** Average algal dry mass and volatile solids for carbon dioxide (blue) and control (open) treatments. Error bars represent 95% confidence intervals (n = 8). Bars labeled with similar letters were compared statistically and those bearing unique numbers differed statistically (AOV, p < 0.05 for both).

to nitrate, orthophosphate concentrations were reduced by 56% in the first 9 days and declined by a smaller magnitude in the second half of the experiment (Fig 5B, Bonf, p < 0.001 for both). Total nitrogen declined by 90% over the course of the experiment (Fig 5C, RM, p < 0.001). Similarly, floway systems averaged 89% total phosphorus removal by day 18 (Fig 5D, RM, p < 0.001).

This study also characterized the statistical significance of the interaction between time and treatment on nutrient concentrations. Significant treatment by date interactions were observed for nitrate (Fig 5A, RM, p < 0.001) and total phosphorus (Fig 5D, RM, p = 0.032), but no interactions existed for orthophosphate or total nitrogen (Fig 5B & 5C, RM, p > 0.11 for both). Treated floways removed three times more nitrate from days 1 to 9 compared to controls. However, by the end of the experiment, nitrate concentrations were equivalent among carbon-infused and control floways. Carbon dioxide reduced total phosphorus concentrations by an average of 93.8% compared to only 83% reduction in controls (day 1 to 18).

## Experiment 2: pH manipulation

To assess the effects of pH on algal productivity, treated and control floways were analyzed for algal biomass growth. Dry algal mass differed significantly among pH treatments (Fig 6, AOV,

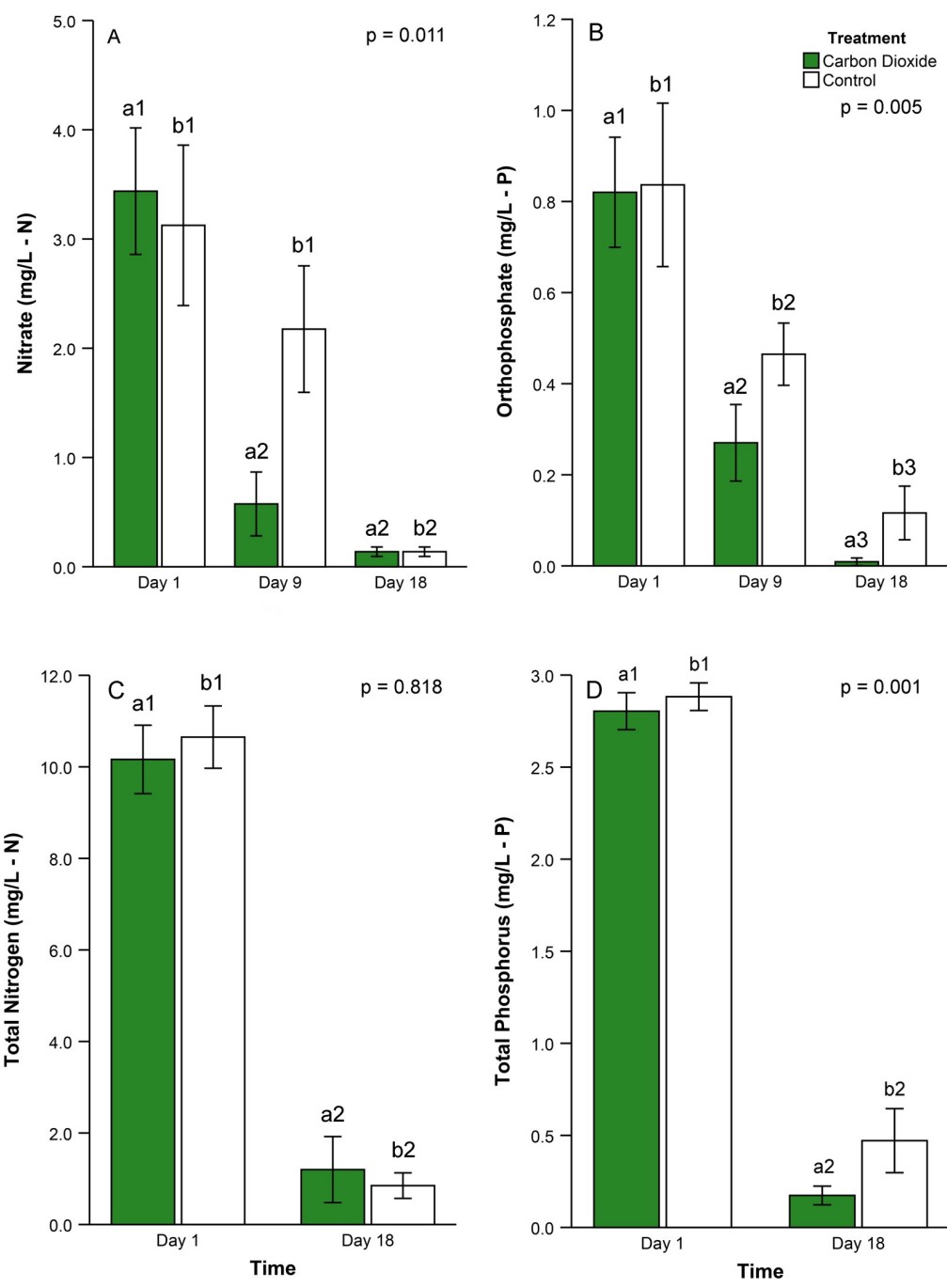

**Fig 5. Nutrient concentrations in carbon dioxide infusion experiment.** Average (A) nitrate, (B) orthophosphate, (C) total nitrogen, (D) total phosphorus concentrations (mg/L) in wastewater at different time periods in the carbon dioxide experiment (n = 24). Error bars represent the 95% confidence intervals. Bars labeled with similar letters were compared statistically and those labeled with unique numbers differed statistically in pairwise, post hoc comparisons (Bonf, p < 0.05 for all significant comparisons). Probability indicates the p-value associated with the overall treatment effect (i.e., carbon dioxide vs control) from the repeated measures statistic.

p = 0.005). Acid treated floways had 40% less algal dry mass compared to controls (Tukey, p = 0.045), while controls had only 14% lower algal dry mass compared to neutralized floways (Fig 6, Tukey, p = 0.549).

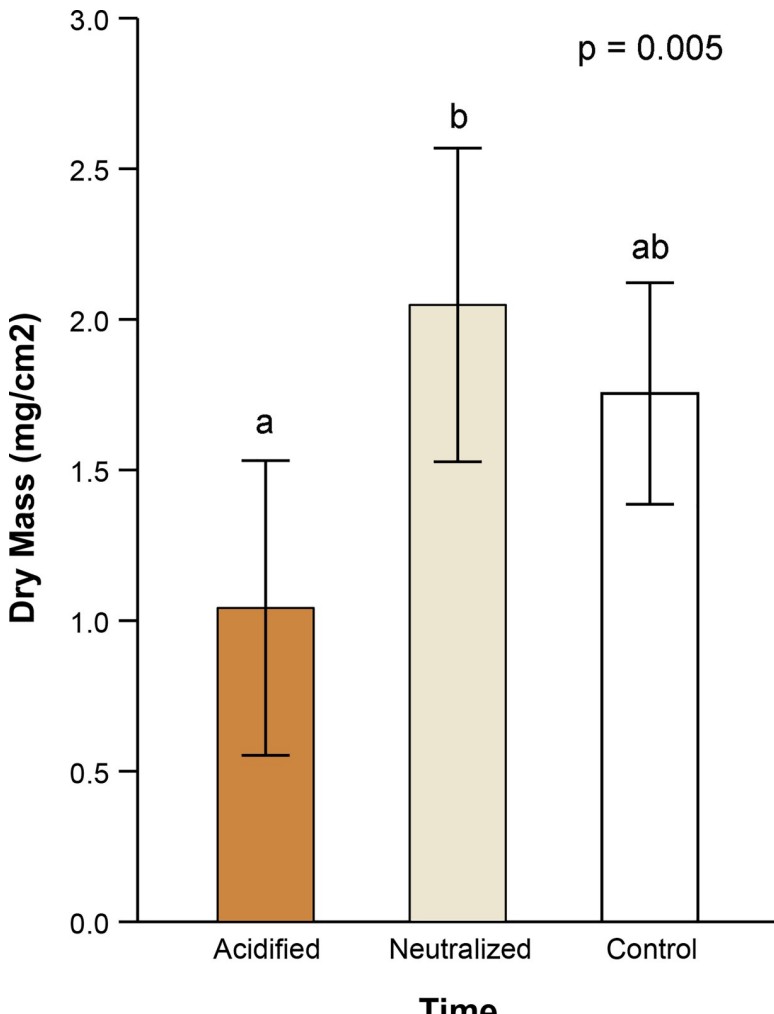

**Fig 6. Effects of acid treatment on algal dry mass.** The average algal dry mass for floways treated with hydrochloric acid, neutralized hydrochloric acid, or nothing (i.e., control). Error bars represent 95% confidence intervals. Bars not sharing the same letter differ statistically in pairwise, post hoc comparisons (Bonf, p < 0.05 for all significant comparisons). Probability indicates the p-value associated with the overall treatment effect (i.e., acidification regime) from the repeated measures statistic.

To more completely assess the effects of pH on floway performance, treatment effects on nutrient concentrations were also quantified in this experiment. Nitrate differed significantly among pH treatments (Fig 7A, RM, p = 0.043). Nitrate concentrations in neutralized treatments were 8% lower than acidified treatments (Tukey, p = 0.033). However, orthophosphate (Fig 7B, RM, p = 0.12), total nitrogen (Fig 7C, RM p = 0.09), and total phosphorus concentrations (Fig 7D, RM p = 0.37) were statistically indistinguishable among treatments.

All nutrient concentrations (i.e., nitrate, orthophosphate, total nitrogen, total phosphorus) declined significantly across time in the pH experiments (RM, p < 0.001 for all). Over the course of the experiment, nitrate concentrations declined significantly by an average of 94% (Bonf, p < 0.001); 62% of the nitrate reduction occurred in the final 9 days (Fig 7A, Bonf, p < 0.001). Floway orthophosphate concentrations showed an average of 81% reduction during this experiment. In contrast to nitrate, most of that reduction (65%) occurred in the first 9 days (Fig 7B, Bonf, p < 0.001 for both). Total nitrogen concentrations were reduced by 89%

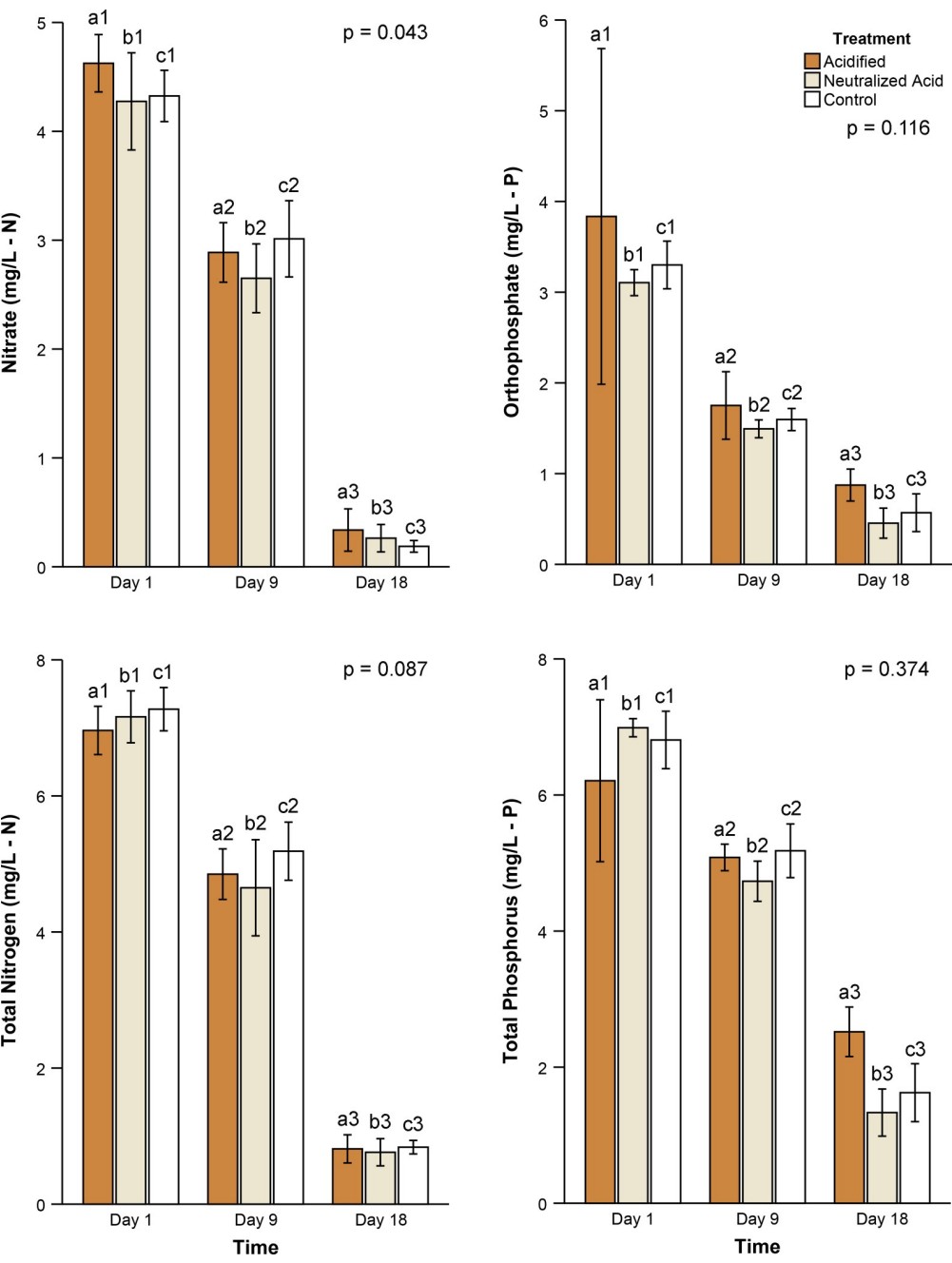

**Fig 7. Acid addition effects on nutrient concentrations.** Average (A) nitrate, (B) orthophosphate, (C) total nitrogen, (D) total phosphorus concentrations (mg/L) in acidified (orange), neutralized (tan), and control floways (open). Error bars represent the 95% confidence intervals. Bars labeled with similar letters were compared statistically and those labeled with unique numbers differed statistically in pairwise, post hoc comparisons (Bonf, $p < 0.05$ for all significant comparisons). Probability indicates the repeated measures analysis of variance p-value for the three acidification treatments.

over the 18-day experiment. Like nitrate, 65% of that decline occurred in the last 9 days (Fig 7C, Bonf, $p < 0.001$). Similarly, total phosphorus concentrations declined by 73% during the testing. Two thirds of that diminution occurred in the final 9 days of the experiment (Fig 7D, Bonf, $p < 0.001$ for both).

Total phosphorus showed the only statistically significant interaction between treatment and time (Fig 7D, RM, p = 0.003). This result indicated that the reduction in total phosphorus concentration over time differed among treatments. No significant interaction was found for nitrate (RM, p = 0.3), orthophosphate (RM, p = 0.8), or total nitrogen (RM, p = 0.5). Floways treated with acid, neutralized acid, and controls showed average total phosphorus removal rates of 0.13 mg/L/d, 0.25 mg/L/d, and 0.18 mg/L/d respectively from days 1 to 9. Total phosphorus removal rates increased to 0.28 mg/L/d, 0.38 mg/L/d, and 0.40 mg/L/d respectively in the final 9 days of the experiments.

## Experiment 3: Sodium bicarbonate addition

Adding sodium bicarbonate as an alternative carbon source caused algal dry mass to increase significantly compared to controls (Fig 8, AOV, p < 0.001). Control floways averaged 39% lower algal dry mass compared to those treated with bicarbonate. Volatile solids were on average 71% of dry mass (Fig 8).

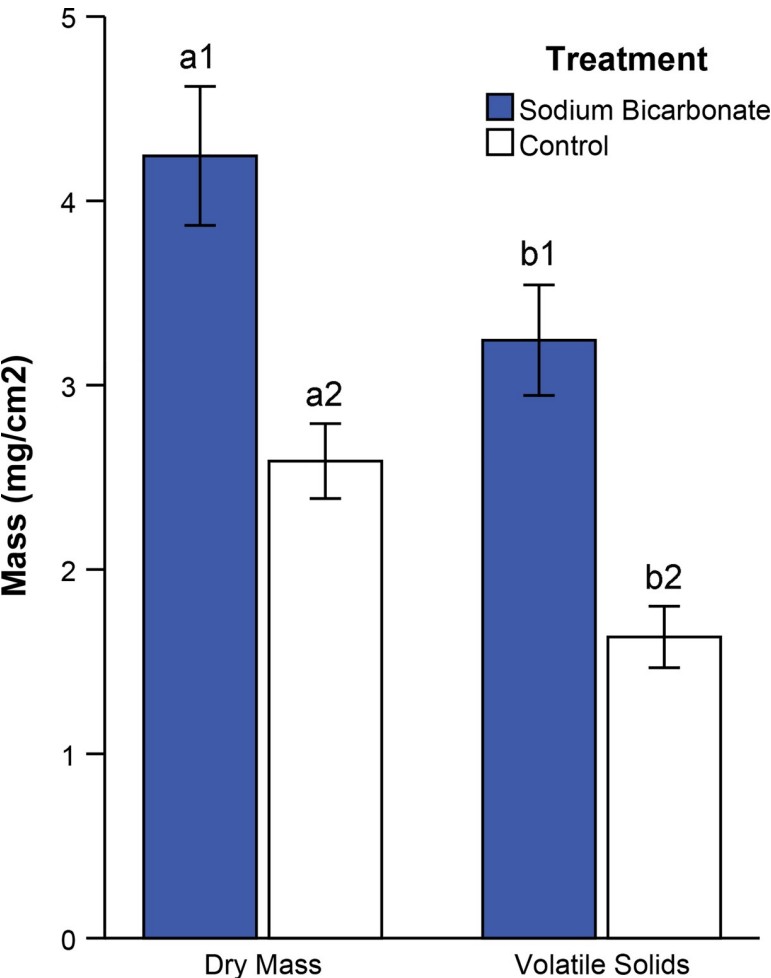

**Fig 8. Effects of sodium bicarbonate on algal mass.** Average algal dry mass and volatile solids for carbon dioxide (blue) and control (open) treatments. Error bars represent 95% confidence intervals (n = 8 for each treatment). Bars labeled with similar letters were compared statistically and those bearing unique numbers differed statistically (AOV, p < 0.05 for both).

In contrast to algal biomass, sodium bicarbonate had no statistically significant effect on nitrate, orthophosphate, total nitrogen or total phosphorus concentrations (Fig 9A–9D, RM, p > 0.22 for all). Over the course of the trials, nitrate composed 29% - 59% of total nitrogen (by mass). Relative to nitrate, orthophosphate was a smaller proportion of the phosphorus pool; it composed only 9% - 17% of total phosphorus.

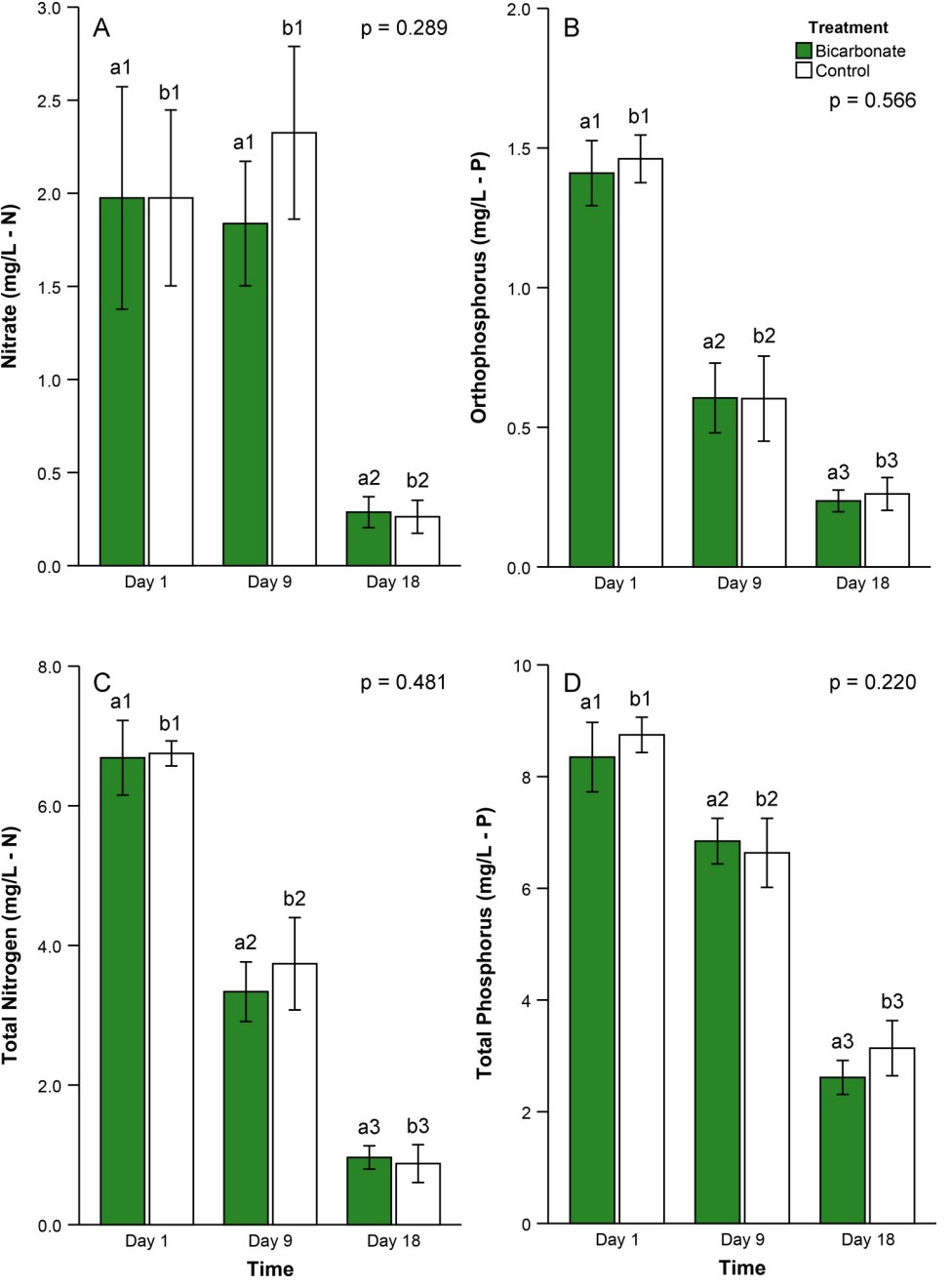

**Fig 9. Effects of sodium bicarbonate on nutrient concentrations.** Average (A) nitrate, (B) orthophosphate, (C) total nitrogen, (D) total phosphorus concentrations (mg/L) in bicarbonate enriched (green) and control floways (open). Error bars represent the 95% confidence intervals (n = 8 for each treatment). Bars labeled with similar letters were compared statistically and those labeled with unique numbers differed statistically in pairwise, post hoc comparisons (Bonf, p < 0.05 for all). Probability indicates the repeated measures analysis of variance p-value for the bicarbonate treatment effect.

Nutrient removal efficiency was analyzed by comparing nutrient concentration over time. Nitrate, orthophosphate, total nitrogen and total phosphorus showed significant declines for all dates tested (Fig 9A–9D, RM, p < 0.001 for all). Nitrate concentrations declined by 86% over the course of the experiment (Bonf, p < 0.001). In contrast to previous experiments, nitrate concentrations remained unchanged in the first half of the experiment (Bonf, p = 1), thus all of the nitrate removal occurred in the final 9 days (Fig 9A, Bonf, p < 0.001). Orthophosphate concentrations were reduced 83% over the 18-day experiment (Bonf, p < 0.001). Most of that reduction (70%) was documented from day 1 to 9 (Fig 9B, Bonf, p < 0.001). Total nitrogen concentrations declined 86% over the course of the experiment with 54% of that decline occurring in the first 9 days (Fig 9C, Bonf, p < 0.001 for both). Concentrations of total phosphorus dropped by 66% during the experiment (Fig 9D, Bonf, p < 0.001). In contrast to total nitrogen, 68% of that change in total phosphorus concentration occurred during the last 9 days (Bonf, p < 0.001).

In the bicarbonate addition experiment there were no significant interactions between treatment and time for any nutrient species measured (RM, p > 0.14 for all). This result indicated that nutrient concentrations within treatments and controls behaved similarly over time.

## Discussion

Algal photosynthetic activity depends on the availability of carbon, nitrogen, and phosphorus [47]. Whereas phosphorus and nitrogen are derived primarily in watershed processes such as runoff, carbon is often supplied by the carbonate-bicarbonate buffer system through diffusion of atmospheric carbon dioxide into water [48]. While the lack of phosphorus and nitrogen can impede algal productivity in oligotrophic freshwater lakes [49], less is known about conditions that cause carbon availability to limit photosynthesis [48]. Theoretically, during periods of rapid algal growth (i.e., high light and nutrient availability), algal uptake of dissolved inorganic carbon during photosynthesis could deplete local sources of carbon faster than diffusion can replenish it [48]. Thus, carbon could become the limiting factor in algal productivity in nutrient rich waters such as those found in tertiary algal wastewater treatment systems.

This study examined carbon limitation in a series of wastewater floway experiments. Results showed that infusing gaseous carbon dioxide into wastewater enhanced dry algal biomass by 66%. This finding is consistent with the hypothesis that algal productivity can become carbon limited during heightened states of photosynthesis. Cole et al. [50] reported increased algal biomass (~40%) when they infused gaseous carbon dioxide into circulating phytoplankton dominated wastewater ponds. Similarly, Park et al. (2011) [51] infused wastewater-filled high rate algal ponds with gaseous carbon dioxide which resulted in a 30% increase in micro-algal productivity. Woertz et al. [33] recorded a 38% increase in algal productivity when carbon dioxide was added to the wastewater in stirred 1L bottles. Similar biomass improvements (39%) were reported in lab-scale algal membrane bioreactors growing the green alga *Chlorella* in secondary wastewater [52]. Furthermore, laboratory bioreactors inoculated with *Scenedesmus obliquus* growing on a wastewater/culture mix showed dissolved inorganic carbon limitation when light was readily available [53]. Biomass increases are not the only effect that might be expected when algae productivity is stimulated with carbon dioxide.

To characterize other important effects of carbon dioxide infusion on algal wastewater treatment systems, this study analyzed nutrient concentration changes and calculated nutrient removal rates. The addition of carbon dioxide (relative to controls) resulted in greater removal rates of total phosphorus concentrations but not nitrate, total nitrogen, or orthophosphate. Sutherland et al. [25] observed a decrease in nitrogen removal when adding carbon dioxide to wastewater in high rate algal ponds. Mulbry et al. [23] found no significant difference in

nutrient removal when carbon dioxide was supplemented in dairy effluent. Wastewater infused with carbon dioxide in the high rate algal ponds of the Park and Craggs [54] experienced a 17% decrease in nitrogen removal which they concluded was the result of reduced ammonia volatilization. Ammonia volatilization has been shown to rise as the pH of wastewater increases [51]. Thus carbon dioxide infusions, which acidify wastewater, can cause a reduction in ammonia volatilization and thus reduce overall nitrogen removal.

The magnitude of ammonia volatilization is also dependent on the type of wastewater being treated. Secondarily treated municipal wastewater has nitrogen primarily in the form of nitrate (2.0–4.4 $NO_3$ –N mg/L) rather than ammonia (0.4 ± 0.1 $NH_4$ –N mg/L, mean ± SD from Columbus Water Works, Inc. December 2016 –December 2018). In contrast primary treated municipal wastewater, used in the Sutherland et al. [25] and Park et al. [51], has nitrogen primarily as ammonia or ammonium (~55 $NH_3$ –N mg/L, ~60 $NH_4$–N mg/L respectively). Because this experiment studied secondarily treated wastewater, there existed lower ammonia concentrations and little likelihood of ammonia volatilization. Thus, nitrogen removal from these algal floways systems was from mechanisms other than ammonia volatilization (e.g., algal uptake or denitrification).

Gaseous carbon dioxide can reduce orthophosphate removal >99% when it is infused into wastewater with multi-species algae cultures [33]. Similarly, orthophosphate and total phosphorus concentrations have been reduced by 93% and 83% respectively in algal turf scrubbers™ treating nutrient enriched water from agricultural canals [55]. The present study found no difference in orthophosphate removal rate in floways infused with gaseous carbon dioxide and controls. In contrast, total phosphorus concentrations were reduced by 94% and 83% in our carbon dioxide treated and controlled floways respectively.

Because the addition of carbon dioxide to wastewater effects both pH and carbon content of the water, most experiments, including experiment one reported here, are confounded by a second uncontrolled variable. To better isolate the effects of carbon limitation versus pH inhibition on algal productivity in wastewater floways, multiple experiments need to be conducted to test each variable separately.

During photosynthesis, algae uptake local carbon from the water in the form of carbon dioxide (at low pH values only) and bicarbonate [56]. Active photosynthesis causes the pH of the surrounding water to shift the dominate form of carbon to carbonate [21]. Because it is difficult for most species of freshwater algae to uptake carbon in the form of carbonate [48], algal productivity could be limited during elevated pH conditions typical of algal treatment systems. During the carbon infusion experiment, carbon dioxide maintained the pH of the wastewater at pH = 6.45 whereas controls averaged pH = 9.25 by the end of the experiment. This difference in pH and carbon availability during carbon dioxide infusions confounds the interpretation of the experiment's results. To evaluate the importance of this confounding pH effect, a second experiment was designed to control pH using carbon-free acids (hydrochloric acid) to see if pH regulation alone controls algal productivity in wastewater floways.

The second experiment was maintained at a near-constant neutral pH (6.45) using hydrochloric acid additions. Counter to expectations, the acidification treatment resulted in a 51% decrease in algal biomass compared to controls. This finding did not support the hypothesis in the strong inference framework that pH control improves algal productivity. Azov [30] reported a 65% increase in algal biomass for *Scenedesmus obliquus* and a 95% increase in *Chlorella vulgaris* when carbon dioxide was used to maintain algal cultures at a pH of 7. Although Azov attributed the increase in biomass to pH regulation, this finding is possibly due to an increase in the availability of dissolved inorganic carbon in the infusion treatments [32]. *Oedogonium*, a green filamentous algae, grown in tumble cultures infused with flue gas (pH between 8.5 and 7.5) grew 40% more biomass compared to controls [50]. In contrast, Moheimani [29]

observed no change in *Tetraselmis sueccica* biomass (i.e., microalga) compared to controls when carbon dioxide was used to regulate the pH of the wastewater inside six-liter glass vessels. The biomass of both *Thalassiosira pseudonana* and *Thalassiosira oceanica* diatoms in batch cultures have been reported to decrease by 10% at pH 9.4 compared to carbon dioxide infused cultures with a pH of 7.9 [28]. The lower biomass at high pH was hypothesized to result from a decrease in available inorganic carbon (e.g., carbon dioxide and bicarbonate) as pH levels increased [28].

The nutrient removal results in our pH experiment did not match our expectations. The addition of hydrochloric acid in this pH experiment decreased nitrate removal compared to the neutralized acid controls. Regulation of pH with acid had no detectable effect on total nitrogen, orthophosphate or total phosphorus removal. In contrast to our findings, *Chlorella vulgaris* grown in cultures with bacteria (*Bacillus licheniformis*) had greater nitrogen removal (86%) when pH was maintained at 7 (with sodium hydroxide) compared to unregulated controls (78%). This result was attributed to higher algal biomass in pH regulated treatments [57].

To more directly examine the importance of dissolved inorganic carbon limitation for algae, a third experiment was conducted that tested algal responses to an alternative carbon source, sodium bicarbonate, which has a limited effect on wastewater pH. This study found that controls had 39% lower dry algal mass compared to sodium bicarbonate treated floways. Final mean alkalinities for sodium bicarbonate and control floways were 804 mg/L as $CaCO_3$ and 217 mg/L as $CaCO_3$ respectively. These measurements indicated that sodium bicarbonate increased dissolved inorganic carbon by more than 3-fold. Alkalinity remained relatively constant for treated and control floways throughout the experiment. It is important to note that there was no significant difference in pH between treated and control floways. The results of this experiment are consistent with the hypothesis that during heightened states of photosynthesis, algae become carbon limited.

Although not as effective, sodium bicarbonate can be used as an alternative to carbon dioxide as a source of dissolved inorganic carbon for algae [58]. These findings are possibly due to differences among algal species in their affinity for carbon dioxide versus bicarbonate, since bicarbonate can be more difficult to uptake [48]. However, White et al. [42] reported that sodium bicarbonate additions at 1 g/L resulted in a 90% increase in algae biomass in cultures of the microalgae *Tetraselmis sueccica* and *Nonnochloropsis salina*.

The additional algal biomass in the sodium bicarbonate treatments was expected to enhance nutrient removal, however, bicarbonate had no significant effect on nitrogen or phosphorus removal compared to controls. After day 9 of the experiment it was discovered that a few midge larvae had invaded the control floways and were consuming algae. It is unclear how these grazers would affect nutrient removal from the water column. The absence of a treatment effect on nutrient removal is particularly perplexing given that biomass production rates were very similar between experiments; experiment 1 showed strong nutrient removal rates (Fig 5) while experiment 3 had lower rates of nutrient removal (Fig 9).

The biological composition of these artificial ecosystems could be an important determinant of their treatment effectiveness, biomass production, or biomass-to-energy conversion efficiency. Measurements revealed the periphyton communities had relatively high volatile solids (70+ %). *Oedegonium*, one of the principle species in these experiments, has high productivities [59] and is composed of a high proportion of carbon [60]. This rich organic composition improves potential for conversion of the algae into biogas using anaerobic biodigestion, particularly if mixed with sewage solids [61]. The economic feasibility of filamentous wastewater treatment systems is greatly improved when the biomass can be effectively converted to energy rich biofuels [17].

## Conclusion

While several studies have reported that carbon dioxide infusions in algal wastewater treatment systems stimulate algal productivity [33, 51, 50, 52], few studies have identified the causal mechanism [see 31 for exception]. This study confirmed earlier experiments showing that carbon dioxide stimulates algal productivity in periphytic algae wastewater treatment systems. Furthermore, the experiments show inconvertible evidence that the mechanism causing this effect is carbon-limitation rather than pH moderation. Experiments revealed that maintaining a constant, near-neutral pH using HCl had no effect on algal biomass or nutrient removal and thus was inconsistent with the hypothesis that pH moderation controls algal growth in wastewater treatment systems. Furthermore, experiments showed that adding a dissolved inorganic carbon source, sodium bicarbonate, stimulated algal biomass comparable in magnitude to that of carbon dioxide.

The reduced availability of dissolved inorganic carbon may be a result of heightened states of photosynthesis in nutrient enriched wastewater treatment systems. In these systems algae rapidly remove usable carbon and cause a shift in the buffer system to relatively bio-unavailable carbonate as the pH increases. This study showed that the addition of carbon dioxide or bicarbonate replenishes the dissolved inorganic carbon needed to maintain the productivity of filamentous algae.

These findings emphasize the value of injecting carbon into algal treatment floways. Carbon augmentation can improve treatment efficiency by increasing nutrient uptake, growing more biomass for biofuel, and shrinking the size of the system's physical layout. Carbon enrichment could make algal wastewater treatment more economically viable and reduce the need for utilities to invest in costly and less sustainable alternative technologies for nutrient removal. Algae could also be harvested more frequently and produce greater amounts of biomass for biofuel production [51]. For wastewater treatment facilities that contain bioreactors, the source of carbon dioxide could come from gases scrubbed from the anaerobically produced biogas. Thus, algal treatment floways could be used to improve carbon capture. This study's results confirm that carbon augmentation has the potential to improve the sustainability of nutrient recovery during wastewater treatment. Technologies that implement natural processes, such as algal wastewater treatment floways, to reverse the effects of anthropogenic influence on the environment stand to lead the way in natural resource management and remediation. The widespread adoption of algal treatment technologies may help solve the growing global challenge of nutrient runoff and its resulting eutrophication of our fresh and estuarine ecosystems [62].

## Acknowledgments

The authors acknowledge the contributions of many students at Columbus State University (CSU). Caroline Humphries (CSU) operated and sampled the experimental systems throughout the project. James "Will" Kiourtsis (CSU) and Care Bacon (CSU) helped construct the research floways. Mengyuan Li (CSU) provided support during initial testing. William Kent (Columbus Water Works, Inc.) provided permissions for us to access wastewater from the South Columbus Water Resources Facility. Cliff Ruehl (CSU-Biology) and Stacey Blersch (CSU-Earth and Space Sciences) lent their expertise during various phases of this project. Rex Lowe (Bowling Green State University-Biology [emeritus]) provided visual confirmation of dominant algal taxa.

## Author Contributions

**Conceptualization:** Troy A. Keller.

**Data curation:** Brandon J. Furnish, Troy A. Keller.

**Formal analysis:** Brandon J. Furnish, Troy A. Keller.

**Funding acquisition:** Brandon J. Furnish, Troy A. Keller.

**Investigation:** Brandon J. Furnish.

**Methodology:** Brandon J. Furnish, Troy A. Keller.

**Project administration:** Brandon J. Furnish, Troy A. Keller.

**Supervision:** Troy A. Keller.

**Visualization:** Brandon J. Furnish.

**Writing – original draft:** Brandon J. Furnish.

**Writing – review & editing:** Troy A. Keller.

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
