## [Decision Letter · Decision Letter 0]

23 Jul 2020

PONE-D-20-18899

Carbon limitation in hypereutrophic, periphytic algal wastewater treatment systems

PLOS ONE

Dear Dr. Keller,

Thank you for submitting your manuscript to PLOS ONE. After careful consideration, we feel that it has merit but does not fully meet PLOS ONE’s publication criteria as it currently stands. Therefore, we invite you to submit a revised version of the manuscript that addresses the points raised during the review process.

We look forward to receiving your revised manuscript.

Kind regards,

Leonidas Matsakas

Academic Editor

PLOS ONE

Journal Requirements:

'This research was partial funded by Columbus State University SRACE grant (BJF) and the Department of Earth and Space Sciences (TAK). The funding groups did not influence the research conclusions or its publication.'

We note that one or more of the authors are employed by a commercial company: Liberty Utilities

3. Please upload a copy of Figure 11, to which you refer in your text on page 23. If the figure is no longer to be included as part of the submission please remove all reference to it within the text.

Reviewers' comments:

Reviewer's Responses to Questions

**Comments to the Author**

1. Is the manuscript technically sound, and do the data support the conclusions?

Reviewer #1: Partly

Reviewer #2: Yes

2. Has the statistical analysis been performed appropriately and rigorously? 

Reviewer #1: No

Reviewer #2: Yes

3. Have the authors made all data underlying the findings in their manuscript fully available?

Reviewer #1: No

Reviewer #2: Yes

4. Is the manuscript presented in an intelligible fashion and written in standard English?

Reviewer #1: No

Reviewer #2: No

5. Review Comments to the Author

Reviewer #1: Major comments

-I found ms is ordinary and not properly organized. Needs to provide more data in the form of figures and tables.

-No statistical figures are given in figures, also no legends provided.

-Information about algal species composition not given.

Reviewer #2: The paper presents an important topic that deserves to be better understood. So the theme is relevant.

Some necessary corrections and suggestions:

Review the units, avoid M and N, use mol L-1 for both.

line 207 – please, use… author (36)

line 238 - what are the other experiments?

The title of the figures looks like a statement and not a title, please review them all.

There is no determination of the trophic level of the experiments. I believe it could enrich the discussion.

line 269 - Why does this happen? "On average, the algae were composed of 87% volatile solids by mass"

I noticed that several old references were cited. Very few in the last 5 years and the problem studied is the focus of many current researches, discussing the environmental impact and methods of remediation. Please, update your references.

6. PLOS authors have the option to publish the peer review history of their article (what does this mean?). If published, this will include your full peer review and any attached files.

Reviewer #1: No

Reviewer #2: No

---

## [Author Response · Author response to Decision Letter 0]

6 Sep 2020

I made the following changes:

• Reformatted the manuscript following the templates provided.

• Updated the Financial Disclosure and the Competing Interests sections

1. Financial Disclosure: “This research was partially funded by Columbus State University’s SRACE grant [BJF], Liberty Utilities (BJF), and the Department of Earth and Space Sciences [TAK]. The funding groups did not influence the research conclusions or its publication. Liberty Utilities provided support in the form of a salary for one author [BJF], but this entity did not have any additional role in the study design, data collection and analysis, decision to publish, or preparation of the manuscript. The specific roles of the authors are articulated in the ‘author contributions’ section.

2. Competing Interests: Liberty Utilities is a commercial entity that employed and paid the salary of one author [BJF] during the final preparation of this manuscript. This does not alter our adherence to PLOS ONE policies on sharing data and materials. The authors declare that no competing interests exist. 

• Is the manuscript technically sound, and do the data support the conclusions?

1. This study reports on a concerted effort to examine the scientific question of whether carbon limitation exists in nutrient over-enriched, filamentous algal treatment systems using three independent experiments. Each experiment was conducted with 8 replicates for the controls and the experimental units. The experiments were conducted in laboratory conditions to reduce environmental variability and ensure that the results were robust and repeatable. The conclusions were drawn directly from the results. For these reasons we agree with Reviewer #2 that this manuscript represents high quality science deserving of publication in PLOS ONE.

• Has the statistical analysis been performed appropriately and rigorously? 

1. The data gathered in this paper were analyzed using the most appropriate statistical models available. In many cases the dependent variables (e.g., nutrient concentrations, pH, or temperature) were measured repeatedly in the same floway. This repetition violates the assumption of independence required by most analysis of variance statistical models. Therefore, where appropriate, the authors used the repeated measures analysis of variance which directly accounts for this lack of independence. Further, the authors analyzed the validity of the repeated model assumptions (e.g., sphericity) and only reported Greenhouse-Geisser corrected significance values. While reviewer #1 indicated that the statistical analysis was not appropriate or rigorous, they provided no commentary to justify their conclusion. Reviewer #2 found our analysis acceptable. Given these conflicting reviews (and a paucity of written feedback), we have chosen to keep the statistical analysis as reported, however we have revised the statistical analysis section of the methods to better clarify and justify our approach. Furthermore, we strengthened the justification for our approach by citing additional references and added statistical test results to the figures. 

• Have the authors made all data underlying the findings in their manuscript fully available? 

1. Here again the reviewers disagreed. Regardless, the raw data did exist online prior to submission and continues to exist in a publicly accessible website hosted by our university. To address Reviewer #1’s concerns, we have uploaded all of the descriptive statistics, statistical tables, and raw data to that website. The datasets can be found at https://csuepress.columbusstate.edu/datasets/2/

• Is the manuscript presented in an intelligible fashion and written in standard English?

1. Both reviewers agreed that the document was not intelligible. We have revised the text to improve the readability of the text. We have also made the specific corrections requested by both reviewers.

• Review Comments to the Authors

1. Reviewer 1 made the following major comments:

i. “Manuscript was ordinary and not properly organized…needs to provide more data in the form of figures” – We expect that revisions to the text, the addition of new citations, and the restructuring of the headings have improved the manuscript’s readability.

ii. “No statistical figures are given in figures, also no legends provided” – We agree with the reviewer and have completely revised all Figs to include statistical results and completely altered the fig titles and legends to address these concerns.

iii. “Information about algal species composition not given” – We acknowledge that this statement is true. While we provided a list of the dominant algal taxa, we do not have data on the relative abundances of the taxa to assess species composition. It was not part our research goal to focus on species composition. Without additional outside funding, we were unable to analyze the species composition in these experiments. 

2. Review #2 made the following specific comments:

i. “Review the units, avoid M and N, use mol L-1 for both” – While we appreciate that there is value in using mol/L, we find that mass per unit volume is the standard for these types of studies. In a quick review of 10 recent publications about algal turf scrubbers (each with 30+ citations), not a single manuscript used mol/L. The reason this field uses mass per unit volume rather than mol per unit volume has to do with the goals of the research. Wastewater treatment technologies are designed to achieve the goal of reducing nutrients or other contaminants. The scientific literature related to this tertiary wastewater treatment has adopted the mass units (e.g., mg/L) to align with guidance established by US regulatory agencies which set industrial treatment load limits, concentration guidelines, and reduction goals in mass units. In this specific case, changes to the units would have no effect on the patterns in the graphs, the results of the statistics, or the conclusions drawn. Thus, we have elected not to change the units to mol/L. 

ii. “line 207 – please, use … author (36)” – Fixed

iii. “line 238 - what are the other experiments?” – Fixed

iv. “The title of the figures looks like a statement and not a title, please review them all.” – We have reviewed and completely revised all figure titles and legends.

v. “There is no determination of the trophic level of the experiments. I believe it could enrich the discussion.” – Given that the experiments were conducted indoors without grazers present the trophic state should have been limited to primary producers. We have revised the text to make this explicit.

vi. “line 269 - Why does this happen? "On average, the algae were composed of 87% volatile solids by mass"” – Oedogonium, one of the dominant species in our experiments, has high volatile solids content (i.e., rich in organic compounds). We have added a new paragraph with several novel references to the discussion in order to specifically to address this important question.

vii. “I noticed that several old references were cited. Very few in the last 5 years and the problem studied is the focus of many current researches, discussing the environmental impact and methods of remediation. Please, update your references.” – We have added several references that were published in the last five years. This raises the total number of recent, research-focused citations (Years 2015-2020) to 12 out of 61 total references (19.6%).

---

## [Decision Letter · Decision Letter 1]

24 Sep 2020

PONE-D-20-18899R1

Carbon limitation in hypereutrophic, periphytic algal wastewater treatment systems

PLOS ONE

Dear Dr. Keller,

Thank you for submitting your manuscript to PLOS ONE. After careful consideration, we feel that it has merit but does not fully meet PLOS ONE’s publication criteria as it currently stands. Therefore, we invite you to submit a revised version of the manuscript that addresses the points raised during the review process.

We look forward to receiving your revised manuscript.

Kind regards,

Leonidas Matsakas

Academic Editor

PLOS ONE

Reviewers' comments:

Reviewer's Responses to Questions

**Comments to the Author**

1. If the authors have adequately addressed your comments raised in a previous round of review and you feel that this manuscript is now acceptable for publication, you may indicate that here to bypass the “Comments to the Author” section, enter your conflict of interest statement in the “Confidential to Editor” section, and submit your "Accept" recommendation.

Reviewer #1: All comments have been addressed

Reviewer #2: All comments have been addressed

2. Is the manuscript technically sound, and do the data support the conclusions?

Reviewer #1: Partly

Reviewer #2: Yes

3. Has the statistical analysis been performed appropriately and rigorously? 

Reviewer #1: I Don't Know

Reviewer #2: Yes

4. Have the authors made all data underlying the findings in their manuscript fully available?

Reviewer #1: Yes

Reviewer #2: Yes

5. Is the manuscript presented in an intelligible fashion and written in standard English?

Reviewer #1: Yes

Reviewer #2: Yes

6. Review Comments to the Author

Reviewer #1: MS-PONE-D-20-18899R1

Reviewer is satified with the revised parts but has one major problem relating to stats used:

1. Why different (p) values used in the figures to define the level of significance. Normally p<0.05 used in published ms. I suggest to use similar level of significance for all figures.

Reviewer #2: The paper is better than original manuscript and I consider that can be published. All requests for the initial assessment were made or explained

7. PLOS authors have the option to publish the peer review history of their article (what does this mean?). If published, this will include your full peer review and any attached files.

Reviewer #1: No

Reviewer #2: No

---

## [Author Response · Author response to Decision Letter 1]

26 Sep 2020

I have revised the manuscript to meet the request of the reviewers.

---

## [Editor Report · Decision Letter 2]

29 Sep 2020

Carbon limitation in hypereutrophic, periphytic algal wastewater treatment systems

PONE-D-20-18899R2

Dear Dr. Keller,

We’re pleased to inform you that your manuscript has been judged scientifically suitable for publication and will be formally accepted for publication once it meets all outstanding technical requirements.

Kind regards,

Leonidas Matsakas

Academic Editor

PLOS ONE
---

## [Editor Report · Acceptance letter]

2 Oct 2020

PONE-D-20-18899R2 

Carbon limitation in hypereutrophic, periphytic algal wastewater treatment systems 

Dear Dr. Keller:

I'm pleased to inform you that your manuscript has been deemed suitable for publication in PLOS ONE. Congratulations! Your manuscript is now with our production department. 

Kind regards, 

on behalf of

Dr. Leonidas Matsakas 

Academic Editor

PLOS ONE